# Rethinking Data Selection: The Importance of Coverage over Difficulty in Generative Fine-Tuning

## Abstract

Selecting high-quality training data can reduce computation cost for LLM fine-tuning. Prior data selection methods have developed a variety of scores aiming to reflect what kind of information a data instance can provide to the model, in order to subselect instances for fine-tuning—and a majority of this prior work has focused on scores quantifying difficulty. The intuition in such work is that difficult examples are more informative, and can therefore lead to more efficient fine-tuning. While data selection based on difficulty has shown promise for smaller classification models, in this work we find that such scores are ineffective for fine-tuning LLMs on generative tasks because their narrow focus on "difficult" instances fails to capture the necessary diversity of the input data. We find that in generative tasks, such approaches always fall behind random selection, which our analysis reveals is more representative of the underlying input space—i.e., has better coverage. Motivated by this, we propose a simple clustering-based selection method which selects data that is more representative of the underlying input distribution, enabling selection of smaller subsets of training data for generative tasks. Using a case study on Llama 3 8B (Grattafiori et al., 2024) and OLMo 2 7B (OLMo et al., 2025), we find that the coverage-based approach performs well above difficulty scoring, yielding performance at or above that of random selection across a set of generative tasks.

## 1 Introduction

While scaling—in terms of data, parameters, and compute—has become a standard pathway to success in pre-training LLMs (Hoffmann et al., 2022), it can often conflict with the practicalities (financial or otherwise) of deployment of these models at scale Recent studies have pointed out that, using a small but high-quality dataset can often lead to performance that is competitive with that obtained using a large but potentially noisy dataset (Abbas et al., 2023; Tirumala et al., 2023). Similar observations have been made for instruction tuning (Zhou et al., 2023) and fine-tuning for reasoning (Ye et al., 2025). The field of data selection/pruning, then, presents an alternate pathway to that of data scaling,[1] where the goal is to promote sample-efficient training by selecting a high-quality subset of the available data.

A promising body of research in data selection has led to the development of various "difficulty scores" which leverage the target model's internal signals to rank instances in the training set for level of difficulty. The basic premise is that confidently (or "easily") predicted instances add limited new signal, whereas examples that the model finds "difficult" are more likely to drive meaningful learning (Campbell et al., 2000; Lewis & Gale, 1994; Settles, 2009). As such, difficulty-based data selection has become a popular paradigm in deep-learning, resulting in the proliferation of a variety of difficulty scores (Toneva et al., 2019; Swayamdipta et al., 2020; Paul et al., 2021; Agarwal et al., 2022, i.a.), primarily applied to classification settings, especially in computer vision tasks. In these works, the target model (a classifier) is trained on the most *difficult* or *uncertain* data, with the expectation that these instances are the most informative to learn effective decision boundaries

---

[1]E.g., see Sorscher et al. (2022) for how effective data-pruning can allow models to go beyond power-law scaling and achieve exponential scaling

among classes. While this paradigm has been successful in classification settings, its transfer to LLM fine-tuning is unclear. Unlike in standard classification settings, in which the model's output space is constrained to the fixed label-set, LLMs are typically fine-tuned using the same pretraining task of next-word prediction—such that their output space after fine-tuning is still their entire vocabulary. That is, fine-tuning of LLMs preserves their *generative* nature.

A major reason to hypothesize that difficulty scores may be insufficient for data selection in LLM fine-tuning is that exclusively prioritizing difficult instances results in only capturing a restricted space of the input data, thus suffering from low data coverage. A specific instance of this argument has previously been explored even in case of classification tasks in computer vision (Zheng et al., 2023), who showed that only including difficult instances leads to catastrophic failures at low data-selection percentages, citing lack of sufficient data coverage as a reason. Similarly, efforts in data filtering for pre-training have highlighted the importance of data diversity, and not difficulty, in order to achieve better performance (Tirumala et al., 2023; Sachdeva et al., 2024).

Taking these observations as our motivation, in this paper we explore the extent to which difficulty-based data selection transfers to generative settings. We first catalog five difficulty scores, and extend them beyond their standard usage in fixed-label classification settings.[2] Specifically, we select three tasks for LLM fine-tuning, choosing datasets in which LLMs have notable gaps between zero-shot and fine-tuning performance, and we apply the difficulty scores to perform data selection at various percentages ranging from 1% to 75% of the training data. We then conduct fine-tuning experiments on two separate models. Our results from these experiments suggest that transfer of these selection methods for LLM fine-tuning is indeed very weak: random sampling often yields performance that is on par with or sometimes better than these methods. We then hypothesize that this may be due to the abovementioned limitations of difficulty scores in selecting instances with sufficient *coverage* of the training data. To test this, we first operationalize *coverage* using a method that compares the full data and selected data in terms of distributions over clustering-based partitions of the data. We find that the divergence between the selected data and the full training data strongly correlates with the performance obtained by fine-tuning LMs on the selected data, lending support to our hypothesis. Finally, we devise a purely coverage-based selection method that prioritizes data coverage as operationalized in our analysis. We find that while this method outperforms difficulty-based data selection as well as another recently proposed method that balances between difficulty and diversity (Maharana et al., 2023), it remains on par with random-sampling, only occasionally outperforming it. At the same time, unlike random sampling, our method achieves monotonically increasing performance with respect to the amount of data selected. Overall, while our findings show mixed added value from the clustering based coverage method (relative to random sampling), they robustly support the prioritization of data coverage over difficulty when it comes to LLM fine-tuning.

## 2 RELATED WORK

**Data selection using difficulty** Difficulty-based methods often use artifacts of models' training dynamics to implicitly or explicitly rank training instances by their "difficulty". Swayamdipta et al. (2020) use model confidence to categorize data instances, while gradient based methods like VoG (Agarwal et al., 2022; Anand et al., 2023) measure difficulty using variance of the gradients. Etha-yarajh et al. (2022) use the notion of usable information for a model to define Pointwise $\mathcal{V}$-usable information (PVI), and use it to quantify the difficulty of data instances. All of the above methods have been devised for fine-tuning to perform classification tasks. Difficulty-based selection has also been applied to pre-training and alignment task during post-training: Marion et al. (2023) use perplexity on a large reference model for data selection during pre-training, while Qi et al. (2025) quantify difficulty as the difference between accepted and rejected response reward, and apply the measure to perform data selection during LLM alignment. We build on these prior methods by extending difficulty scores for generative fine-tuning tasks (see Section 3).

**Data selection using diversity** Difficulty-based methods often result in the selection of redundant instances that cover only a limited portion of the data distribution (Settles & Craven, 2008; Xu et al., 2003) Prior work in active learning (Dasgupta & Hsu, 2008; Ash et al., 2020; Sener & Savarese, 2018) has shown that training on diverse data leads to better performance in computer vision tasks.

---

[2]To the best of our knowledge, we are the first ones to do this.

$D^2$ (Maharana et al., 2023) aims to balance diversity as well as hardness for data selection and is primarily focused on fixed-label classification tasks. The instruction-tuning literature (Bukharin et al., 2024; Zhou et al., 2023) has emphasized the importance of data quality as well as data diversity. Our focus is to investigate whether diversity alone is sufficient for selecting data when fine-tuning on generative tasks.

## 3 DIFFICULTY SCORES

We approach our experiments and analyses with the goal of identifying scores that will enable effective subselection of data for generative fine-tuning. We begin by investigating existing difficulty-based scores, which are either derived from the training dynamics of a model trained on the full dataset, or require a single forward pass on the dataset. In this section, we discuss our extension of common difficulty scores to generative fine-tuning.

Common to the data-selection methods used in this work is the notion of a utility function, which determines which instances in the target dataset are selected. Since we are primarily focused on difficulty here, our utility functions will operationalize difficulty in some way, and will prioritize data that are deemed to be more difficult (over those deemed less so). Given a supervised training dataset $D_{train} = \{(x_i, y_i)\}_{i=1}^n$, our aim is to choose a subset $S \subset D_{train}$ such that the model trained on $S$ maximizes accuracy on the test set $D_{test}$. Here $y_i$ represents the output sequence instead of a single label. The utility function $f(x_i) : D_{train} \rightarrow \mathbb{R}$ maps a data instance $x_i$ to a real number representing the difficulty score of the instance. Once we obtain these difficulty scores, we select $S_{m\%}$, or the subset containing the top-$m\%$ of instances ranked by difficulty, as $S_{m\%} = \{x_j \in D_{train} \mid f(x_j) \geq C_{m\%}\}$, where $C_{m\%}$ denotes the $(100 - m)$-th percentile of the difficulty score distribution. Below we describe five utility functions used in our analyses.

**Perplexity**   Language model perplexity is a common, general-purpose measure of an LM's quality at predicting a given sequence distribution, and as such has also been adopted by prior work to quantify the difficulty of data instances with respect to data-selection for pre-training (Marion et al., 2023; Ankner et al., 2024). One benefit of using perplexity as a measure of difficulty for LM fine-tuning is that it can be computed using a simple forward pass over the training data, rather than training the model fully. We adopt perplexity as a difficulty score by selecting instances in the training data with higher perplexity, assuming a correspondence between perplexity and difficulty (i.e., the greater the perplexity on an instance, the worse the model is at predicting it).

**Confidence and Variability**   Swayamdipta et al. (2020) propose two utility functions for classification tasks, both measured across training epochs: (1) a model's confidence of the true label, and (2) its variability. **Confidence** is calculated as the average likelihood of the true label across epochs, while **Variability** is its standard deviation. We extend these measures to generative fine-tuning as follows: let $E$ be the total number of epochs that the LM is fine-tuned for, $x_i$ be the input sequence corresponding to the $i$-th data-instance, and $y_1, \ldots, y_n$ be the ground-truth output sequence, then the confidence ($\mu_i$) and variability ($\sigma_i$) are given as:

$$\mu_i = \frac{1}{E} \sum_{e=1}^{E} \left( \prod_{j=1}^{n} p_e(y_j \mid y_{1:j-1} x_i)^{1/n} \right) \quad \sigma_i = \sqrt{\frac{1}{E} \sum_{e=1}^{E} \left( \prod_{j=1}^{n} p_e(y_j \mid y_{1:j-1} x_i)^{1/n} - \mu_i \right)},$$

where $p_e$ is the LM's sequence probability at the $e^{\text{th}}$ epoch. In order to operationalize difficulty-based selection for these metrics, we use smaller values for confidence, and larger values for variability.

**Variance of Gradients (VoG)**   VoG (Agarwal et al., 2022; Anand et al., 2023) is a utility function based on the idea that easy examples lead to saturated losses and stable gradients early in training, whereas "difficult" examples exhibit gradient variability throughout training. This measure is also originally proposed for classification task, so to apply it to generative fine-tuning we compute the gradients of the logits at the location of target token $y_i$ with respect to the embedding of each word in $x_i$. Averaging the gradients across output length gives us $G_i$. Once $G_i$ is obtained, the computation

of VoG follows exactly the same process as the original VoG method. Selection preference is given to instances with high VoG score. For brevity, we describe VoG score calculation in Appendix A.2

**Pointwise $\mathcal{V}$-usable information (PVI)**    Ethayarajh et al. (2022) propose PVI, which quantifies the usable information an instance provides to a model for predicting the target. To quantify this, PVI involves fine-tuning two copies of the same base model on $D_{train}$ and on a modified dataset $D_\phi = \{(\phi, y_i)|(x_i, y_i) \in D_{train}\}$, where all inputs $(x_i)$ are replaced with an empty string $\phi$. After training, the method obtains two models, $g$ (trained on $D_{train}$), $g'$ (trained on $D_\phi$). Each model can be treated as a function that maps an input to a probability distribution over possible outputs $y$. The PVI for each training instance $(x, y)$ is calculated as $\text{PVI}(x, y) = -\log_2 g[\phi](y) + \log_2 g'[x](y)$. Intuitively, a low PVI means that the input provides less usable information for the model and hence the model struggles to learn from these instances. We apply PVI directly by using the likelihood of the entire output sequence, and select training instances with low PVI as "difficult" instances for fine-tuning.

**Difficulty scores we did not include**    Besides the difficulty scores mentioned above, there are at least two popular scores that we did not include in our experiments. First, the EL2N score (Paul et al., 2021) measures the squared difference between the predicted and (one-hot) true labels for a specific input instance. Second, the *forgetting* score (Toneva et al., 2019) measures the number of times a training instance moves from being classified correctly to being classified incorrectly. We did not include these scores because both of these metrics require multiple fine-tuning runs for stability, as well as a large number of epochs, which is impractical in the context of fine-tuning LLMs.

## 4   EXPERIMENTAL SETUP

To test the effectiveness of each of these data selection metrics in generative fine-tuning, we carry out experiments across 3 target datasets, described below. For all our experiments, we vary the number of selected instances as $\{1, 5, 10, 25, 50, 75\}\%$ of the total training data. We first compute our difficulty scores for the entire dataset (i.e., 100%), use them to select difficult training examples, and then fine-tune a model on the resulting subset. For large-scale model fine-tuning, we use Huggingface Accelerate (Gugger et al., 2022) and utilize Zero3 optimizer in DeepSpeed (Rasley et al., 2020). We compare against a random sampling baseline, in which we randomly sample the given percentage of the training data, and then fine-tune our models on this subset. We repeat this random-sampling with three different random seeds for each selected percentage.

**Models tested**    We experiment on two transformer-based language models, both of which are trained on the standard language model objective: Llama 3 8B (Grattafiori et al., 2024) and OLMo 2 7B (OLMo et al., 2025). We finally fine-tune for 3 epochs using AdamW optimizer with linear scaling and learning rate of `2e-05`. All models use a context window length of 512.

**Datasets**    We choose datasets in which fine-tuning yields marked improvements compared to zero-shot performance for both models (difference of at least 10% points). Based on this criterion, we select three multiple-choice QA datasets: Social IQa (Sap et al., 2019), CommonsenseQA (Talmor et al., 2019), and CosmosQA (Huang et al., 2019) from among all the datasets that we considered (see Table 3 for our initial results from five different datasets for both models). We use the validation set of these datasets in our tests.

**Evaluation**    We follow the MCF (multiple-choice formulation)-based evaluation procedure (Gu et al., 2025). That is, we compute accuracy as the exact-match between the first token of the LM's generated response and the true label. For instance, a generated response to the example in (Table 1) is considered to be correct only if the first token exactly matches the target completion (i.e., answer choice C).

**Difficulty score computation**    While we use only the option label during evaluation, our target ground-truth sequence during difficulty score computation includes both the letter label and the full answer string. That is, had the example in Table 1 appeared in the training set, our difficulty scores would have been computed based on the full target sequence "C. frame the picture". We use the

| Prompt | Question: Kai improved Jan's picture and she loved how it came out. What will Jan want to do next?
A. wanted to be helpful
B. wanted Jan to be glad
C. frame the picture
Answer: |
|--------|------------------------------------------------------------------------------|
| Target | C |

Table 1: Example item from the Social IQA validation set.

full sequence in lieu of only the label since it provides sufficient information for PVI and Perplexity (e.g., in the case of PVI, without the full content, we would be fine-tuning the null model on only the single-letter option labels). Furthermore, this allows us to extend these scores beyond single item predictions to multi-word sequences, thereby enabling their use in long-form context generation, which we leave for future work.

## 5 DIFFICULTY-BASED METHODS ARE OFTEN OUTPERFORMED BY RANDOM SAMPLING

We evaluate the difficulty score selection methods as outlined in Section 4. This helps us in answering our question: Can we rely on difficulty scores to determine which instances are most useful for fine-tuning generative models? Figure 1 shows validation accuracies at the end of training as a function of data selected for each of the five difficulty-based selection methods.

We first see that models fine-tuned with as little as 50% of the full dataset achieve performance highly comparable to models fine-tuned on the entire dataset. Across all models and datasets, the performance curves for most selection methods—including the random baseline—reach a plateau at or before the 50% data mark. This suggests that the datasets considered here contain a significant degree of redundancy, and that data selection methods can be highly beneficial for achieving strong performance with reduced training time and computational cost.

We also find that the most effective data selection method is dependent on the specific dataset. For a given dataset, however, the superior method appears to be consistent regardless of the base model. For example, the variability method consistently yields a top-performing accuracy for the Social IQa dataset on both Llama 3 8B and OLMo 2 7B. Similarly, for both the CosmosQA and CommonsenseQA datasets, VoG consistently provides the highest validation accuracy. At an aggregate level, VoG achieves the highest average accuracy among all difficulty-based methods.

Finally, we find empirically that the random baseline is consistently competitive—and often the best performing method—across all selected percentages and datasets. The surprising effectiveness of this simple baseline motivates our subsequent analysis into why random selection outperforms conventional difficulty-based scoring methods for fine-tuning LLMs.

## 6 DIFFICULTY-BASED SELECTION SUFFERS FROM POOR DATA COVERAGE

Why might simple random sampling be as good as—or even better than—principled, difficulty-based selection methods in these generative fine-tuning settings? A potential explanation lies in the extent to which each method's data-selection scheme "covers" the domain of the dataset. We hypothesize that difficulty-based measures may select for items occupying a smaller, more restricted space within the training dataset, resulting in an unrepresentative sample—whereas a simple random sampling method may cover a larger, more representative space. As a result, a model fine-tuned on a difficulty-based data selection may end up biased toward a smaller space of the task, de-prioritizing features that may be important for tasks that are more general in nature. This hypothesis is consistent with the findings of Zheng et al. (2023) for fixed-label computer vision tasks, described in Section 1.

In this section we test this hypothesis by defining a measure of coverage and testing whether this measure is predictive of how effective a selection method will be. To do so, we must first opera-

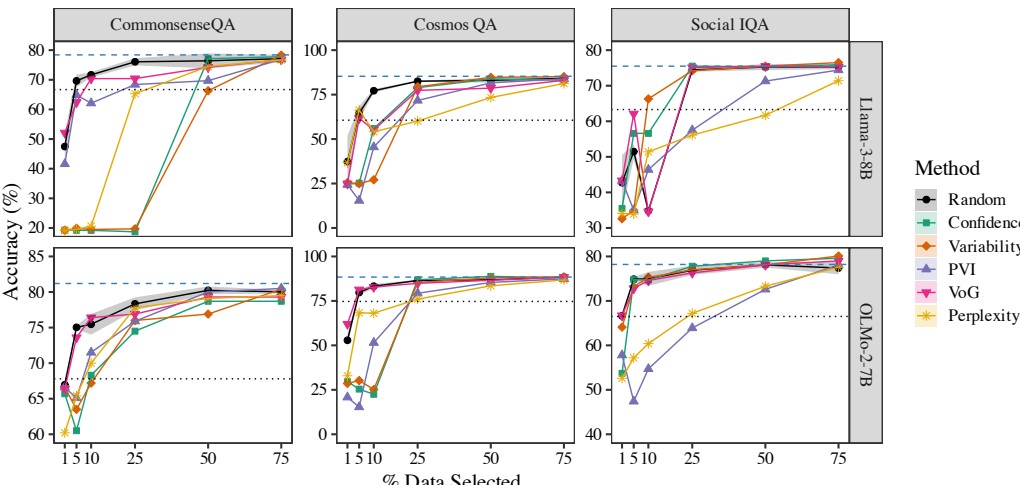

Figure 1: Comparative performance of difficulty-based data selection methods on the Llama 3 8B and OLMo 2 7B models across the three datasets, as a function of the percentage of training data selected. For random selection, the points show the mean test accuracy and shaded regions indicate the $\pm 1$ standard deviation band across runs. The blue dashed line indicates results from fine-tuning on 100% training data, while the gray dotted line indicates zero-shot performance.

tionalize the notion of *coverage*. However, computing how well a given data-selection scheme picks out a diverse, high-coverage sample of the entire training data requires access to the true distribution of the data, which is intractable. In light of this, we rely on a quantized/discretized embedding space, sensitive to the distributional semantic features of the instances in the training set. This part of our method is inspired by a similar component introduced in the computation of MAUVE (Pillutla et al., 2021), a metric for comparing LM-generated data to that of humans, using divergence curves.

Our estimation process is as follows: first, we use the non-fine-tuned state of the target LM to map training data instances $\{(x_1, y_1), \ldots, (x_n, y_n)\}$ to an embedding space $\text{LM}((x_i, y_i)) \to \mathbb{R}^d$, where $d$ is the dimensionality of the embedding space, giving us a collection of vectors $\{\mathbf{v}_1, \ldots, \mathbf{v}_n\}$.[3] We then use k-means clustering (Lloyd, 1982) to quantize these embeddings, allowing us to use the cluster assignments $\{1, \ldots, k\}$ as a support to compare the distribution of the training set to a selected subset. To execute this comparison, we define $P$ and $Q$, the (approximate) distributions of the cluster labels in the full training data and a given selected subset, $S_{m\%}$, respectively, as:

$$P(j) = \frac{1}{n} \sum_{i=1}^{n} \mathbb{1}[\psi(\mathbf{v}_i) = j] \qquad Q(j) = \frac{1}{|S_{m\%}|} \sum_{i=1}^{|S_{m\%}|} \mathbb{1}[\psi(\mathbf{v}_i) = j],$$

where $\psi(\mathbf{v})$ represents the cluster assignment function (i.e., one resulting from the k-means algorithm) that returns a cluster label in $\{1, \ldots, k\}$. We can now compare how different $Q$ is from $P$, by computing the Jensen-Shannon Divergence (JSD) between them:

$$\text{JSD}(P \parallel Q) = \frac{1}{2}\text{KL}(P \parallel M) + \frac{1}{2}\text{KL}(Q \parallel M); \qquad \text{KL}(P \parallel Q) = \sum_{x} P(x) \log \frac{P(x)}{Q(x)},$$

where $M$ is the mixture distribution of $P$ and $Q$, computed as $(P+Q)/2$. Since this measure, as it is currently defined, is sensitive to the chosen value of $k$, we repeat this computation over 10 different random seeds, as well as across a range of values for $k$ per dataset, model, and data percentage. More specifically for $m\%$ selected data, we experiment with values of $k$ ranging from 2 to $mn/100$, incrementing in powers of 2. Overall, selection methods that have lower Average JSDs are more similar to the distribution of the training data.

---

[3]In practice, we use the model's hidden state representation at the last layer and last position as our embedding extractor.

We now explore how this measure patterns against our findings from the previous section, and whether it is a reliable predictor of the effectiveness of different data selection methods. For this, we measure the average JSD for our data selection methods, across different models and datasets. Figure 2 shows the mean rank of each data-selection method, calculated on the basis of its average JSD values across selected-percentages, datasets, and models. We find that the **Random** data selection method is always the one with the lowest average JSD value across all datasets, both models, and all data-selection percentages, suggesting that it systematically achieves the greatest coverage of the training data relative to other methods. Next, we measure the Spearman's correlation between average JSD values of different methods and accuracy obtained as a result of deploying them at each data-selection percentage, across different datasets and models. Insofar as coverage—as operationalized by our metric—is related to data-selection performance, we expect there to be a strong negative correlation between average JSD and accuracy, since methods with poorer coverage (i.e., high divergence) are posited to have poorer performance under this prediction. Table 2 shows our results across models and datasets. In line with the aforementioned prediction, we find strong negative correspondence between average JSD and accuracy, across all model-dataset pairs, with values ranging from -0.71 to -0.90.

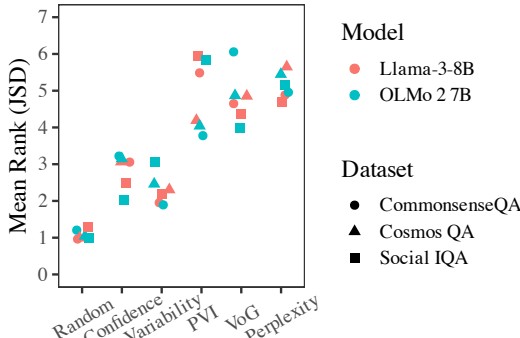

Figure 2: Mean rank (based on Avg. JSD) of each selection method across models and datasets. Methods with lower ranks have lower avg. JSD.

| Model | Dataset | $\rho$ |
|---|---|---|
| Llama 3 8B | CommonsenseQA | -0.71 |
| | CosmosQA | -0.87 |
| | Social IQa | -0.81 |
| OLMo 2 7B | CommonsenseQA | -0.83 |
| | CosmosQA | -0.88 |
| | Social IQa | -0.90 |

Table 2: Spearman's correlation between avg. JSD and accuracy across data-selection percentages and methods, for each dataset and model. Lower values indicate greater correspondence between coverage and performance.

Our results suggest that this notion of coverage may explain the weakness of the difficulty scores for selecting subsets of training data: we find that reduced coverage directly correlates with lower task accuracy for models trained on these subsets. Random sampling preserves coverage, which may be the source of its consistently strong performance in these generative fine-tuning settings.

# 7 A PURELY COVERAGE-BASED DATA SELECTION METHOD

Our analysis in the previous section indicated that selecting subsets with good coverage of the training data might be key to better LLM fine-tuning performance, and may potentially explain the poor performance of difficulty-based methods. We now turn to proposing a method that exclusively prioritizes coverage-based selection, to shed further light on the extent to which coverage alone can serve as an alternative to difficulty-based selection for fine-tuning LMs.

Our method builds directly from our analysis in the previous section. As in our analysis, we use the embedding space of the training data by using the target LM's last hidden state at the last token-position of each training instance. Then we cluster these embeddings using $k$-means clustering, this time fixing $k$ as $mn/100$, where $n$ is the total size of the training set, and $m$ denotes the data selection percentage. This means that our total number of clusters is the same as the number of points to be selected. We then simply sample one instance from each cluster, allowing all cluster points to be represented, thereby encouraging "good" coverage of the data.[4] We then perform standard fine-tuning as described in Section 4 to obtain performance estimates that represent our coverage-based selection method. To capture randomness introduced by both our clustering step as well as random

---

[4]We explored sampling using difficulty measures such as perplexity, etc. but our initial results in this direction were consistently worse than random sampling.

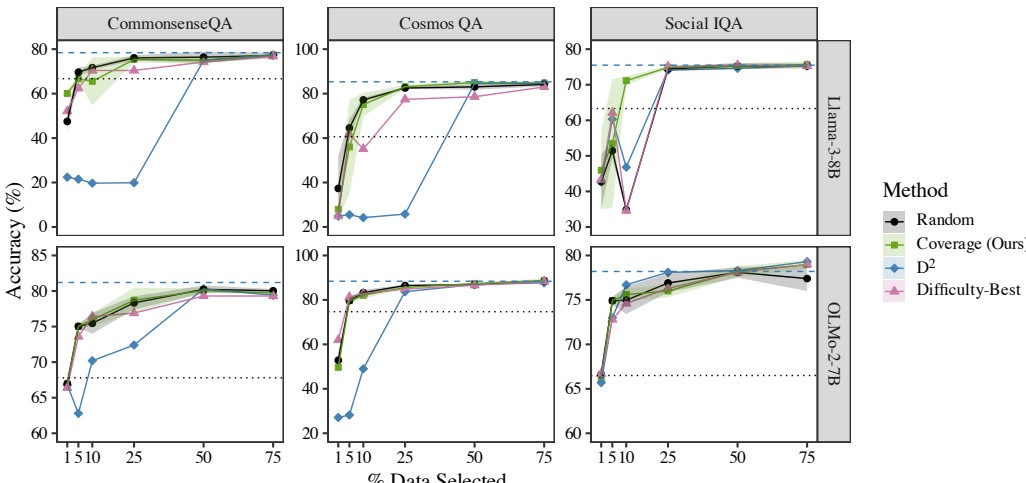

Figure 3: Performance of the Coverage-based selection method on Llama 3 8B and OLMo 2 7B models as a function of training data selected, along with comparisons to random selection, he best overall difficulty-based method (VoG), and $D^2$. For random selection and Coverage, the points show the mean test accuracy and shaded regions indicate the $\pm 1$ standard deviation band across runs. The blue dashed line indicates results from fine-tuning on 100% training data, while the gray dotted line indicates zero-shot performance.

sampling, we repeat this experiment three times for each dataset, model, and selected percentage, with different random seeds. We would like to point out that unlike previous works like (Tirumala et al., 2023; Sorscher et al., 2023) we do not rely on pretrained embeddings. We use the target model's embeddings since these may provide a more direct signal of data usefulness.

We compare performance obtained by our method (averaged over three random seeds) to those from random sampling, as well as the best (overall) difficulty-based selection method—i.e., VoG. In addition, we also compare against $\mathbf{D^2}$ **Pruning** (Maharana et al., 2023, abbreviated as $D^2$). $D^2$ aims to balance difficulty and diversity by representing the training set as a graph, where each instance is a node initialized with a difficulty score, and edges are weighted by the distance between nodes in embedding space. The forward message-passing step propagates node features so that each instance's score is adjusted by the difficulty of its neighborhood, thereby promoting diversity across regions of the data distribution. Instances are then ranked by these updated scores in the reverse message-passing step. In our adaptation, we use the variability scores from Section 3 as node features, following Maharana et al. (2023), and the target LLM's final hidden state at the last token position as embeddings.

Each comparison here meaningfully addresses aspects of our research questions: 1) by comparing against the best difficulty-based method, we directly pit diversity against difficulty; 2) by comparing against $D^2$ Pruning, we can shed light on how well data-diversity (captured by coverage) alone fares against a selection scheme that balances between the two extremes; and 3) by comparing against random, we test if a more principled and stable method of data-selection can compete against the uncertainty of exclusive random-sampling.

**Results** Figure 3 compares results using our Coverage method to those from random sampling, the best difficulty-based method (VoG), and $D^2$. First, we see that accuracies obtained from using $D^2$ are consistently worse at lower percentages, across both models for CommonsenseQA and CosmosQA. In fact, $D^2$ seems to under perform even VoG in most cases, suggesting that balancing between difficulty and diversity in this setting harms rather than helps with data selection. There are certain instances where $D^2$ is in fact the best performing method (e.g., Social IQa) but these are overshadowed by the fact that it performs substantially worse than zero-shot performance for other datasets. While VoG is fairly competitive against random and our coverage method, there are cases where it is clearly outperformed by them (e.g., CosmosQA and Social IQa with Llama 3 8B), and

it sometimes shows a non-monotonic trajectory with respect to the percentage of data selected. In contrast, our coverage-based method almost always shows an upward trajectory with respect to the selected percentage, obtaining similar performance to the fully fine-tuned model at even 25% data. Overall, except in a small number of instances, the finding from Section 5 seems to persist—random sampling is rarely systematically outperformed by any more principled method tested so far.

## 8 DISCUSSION

Our results suggest that coverage rather than difficulty should be prioritized when doing data selection for generative fine-tuning. This is indicated by the observation that random selection and our coverage-based method perform better than difficulty-based scores in generative fine-tuning tasks.

Why do we find that lack of coverage leads to such weak performance in generative fine-tuning? Diversity has been shown to be a driver of performance during pretraining (Tirumala et al., 2023; Sachdeva et al., 2024), so we postulate that the particular importance we see here for coverage in this setting is attributable to the greater similarity between generative fine-tuning and more generalized language modeling, by contrast to the more focused decision space of fixed-label classification. More specifically, we speculate that fixed-label classification tasks can be solved by learning a more focused set of items that help to identify the decision boundary, while generative tasks require models to retain more generalized capabilities.

A natural extension of this work would be to combine data difficulty and diversity for data selection. Prior work, such as $D^2$ (Maharana et al., 2023), explores this direction but underperforms both random selection and our coverage-based method, and many times even the best difficulty-based method. We did attempt to combine difficulty and diversity in our method by selecting data with the highest difficulty (e.g., perplexity) from within each cluster—however, as shown in Figure 4 in the Appendix, this approach did not yield any performance gains, suggesting that it was coverage that was driving our observed gains. We leave for future work the investigation of other approaches to combine difficulty and diversity which might benefit data selection in these generative settings.

A practical concern regarding adoption of any of these data selection methods is the compute cost for selection. For instance, almost all difficulty-based scores above, except Perplexity, require training the model on the full dataset—sometimes even twice (in the case of PVI). This is counter-intuitive, since in practical settings if one has the full model trained there would be limited benefit in training another model on a subset of data—in fact, it will result in increased compute costs while also (likely) sacrificing performance. Two exceptions to this are Perplexity and our Coverage method, both of which only require a single forward pass, though this is still a significant amount of compute cost compared to random sampling. We provide the FLOPs utilized for each selection method in Appendix A.3, and a plot of FLOPs vs. accuracy in Figure 5. After random selection, our Coverage method is by far the best in terms of compute efficiency.

**Limitations and Future Work** Our main goal in this work is to investigate the impact of data diversity and data difficulty while fine-tuning for generative tasks. While our empirical analyses reveal a strong correlation between diversity and the task performance, establishing causality is left for future work. Furthermore, while notions of difficulty have been shown to fare well in certain pretraining investigations (Marion et al., 2023), they have not been compared to coverage, leaving open the question of difficulty vs. diversity in those settings. Future work can also explore whether selection utility transfers across model scales, as seen in Bordelon et al. (2023) for learning rates in Vision Transformers, and in Wang et al. (2023) for active learning.

## 9 CONCLUSION

We explored how different difficulty-based data selection methods perform in task-specific fine-tuning settings. We find that each of these methods fall behind random selection. Our results suggest that the strong performance of random selection arises from its better coverage of the training distribution, which appears to have greater importance in generative settings. We also show that a simple coverage-based method consistently outperforms difficulty-based methods, and is at par or sometimes better than random sampling. These findings highlight the likely importance of coverage in data selection for generative fine-tuning.

## 10 REPRODUCIBILITY STATEMENT

Code for all experimental results reported in this paper is provided in the supplementary submission.

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

# A APPENDIX

## A.1 EFFECT OF RANDOM SELECTION VS. PERPLEXITY IN OUR COVERAGE METHOD

Figure 4 shows the impact of using perplexity vs. random sampling for our Coverage method. Using perplexity does not provide any noticeable gain, and in fact hurts performance at lower data-percentages—e.g., see performance on Llama 3 8B for CommonsenseQA, or on OLMo 2 7B for Social IQA.

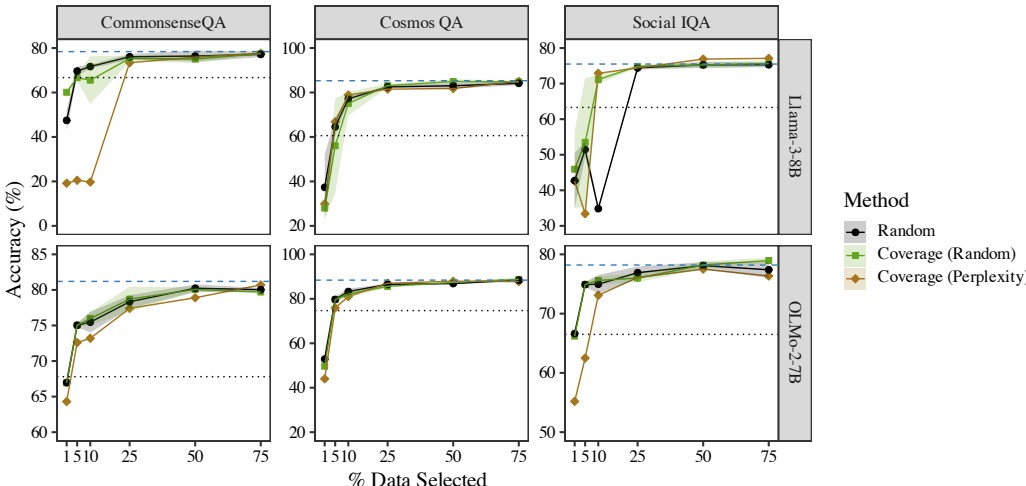

Figure 4: Effect of random sampling vs. selection using perplexity after performing clustering for our coverage-based method.

| Dataset | Model | Zero-shot | 100% |
|---|---|---|---|
| Social IQa | Llama 3 8B | 0.633 | 0.755 |
| | OLMo 2 7B | 0.665 | 0.782 |
| CommonsenseQA | Llama 3 8B | 0.667 | 0.784 |
| | OLMo 2 7B | 0.678 | 0.812 |
| CosmosQA | Llama 3 | 0.606 | 0.853 |
| | OLMo 2 7B | 0.747 | 0.884 |
| OpenBookQA (Mihaylov et al., 2018) | Llama 3 8B | 0.736 | 0.828 |
| | OLMo 2 7B | 0.714 | 0.844 |
| ARC-Hard (Clark et al., 2018) | Llama 3 8B | 0.782 | 0.743 |
| | OLMo 2 7B | 0.709 | 0.799 |

Table 3: Datasets considered for experiments. Zero-shot performance and performance on using the entire train data on Llama 3 and OLMo 2

## A.2 VoG EXTENSION

Let $x_i = x_{i,1}, x_{i,2}, ...x_{i,N}$ and $y_i = y_{i,1}, y_{i,2}, ...y_{i,M}$. Let $A_{i,j}$ be the logit at the location of $j-th$ token in $y_i$. Now, $G_{i,k} = \frac{\sum_{j=1}^{M} \frac{\partial A_{i,j}}{\partial E_{x_{i,k}}}}{M}$.

Here, $G_{i,k}$ represents the gradient of logit at the target location $y_j$ w.r.t the embedding vector of $k-th$ token in $x_i$. We obtain $G_i = G_{i,1}|G_{i,2}|...G_{i,N}$ by concatenating the obtained logit gradients.

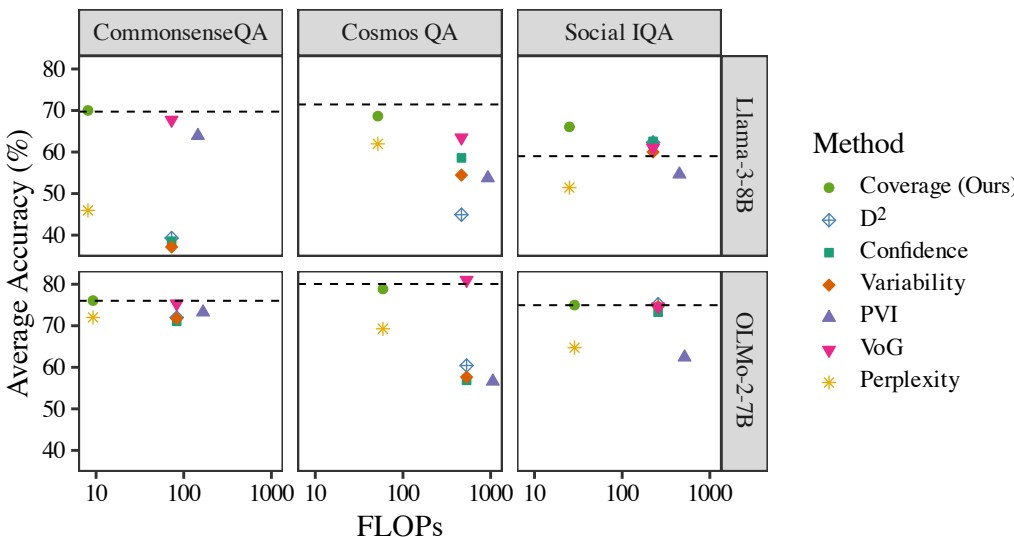

Figure 5: Data Selection FLOPs ($\times 10^{15}$) vs. Average Accuracy (calculated over all percentages) across all methods. X axis is in logarithmic scale. Dashed line indicates Random Selection performance which has a data-selection FLOP value of 0.

We calculate the mean and variance across the epochs. $\mu_i = \frac{\sum_{e=1}^{E} G_i}{E}$

$V_i = \frac{(G_i^e - \mu_i)^2}{\sqrt{E}}$

The (unnormalized) score $v_i$ for each instance $x_i$ is then given by the mean of $V_i$ (that is, we average over the input embeddings)

$VoG(x_i) = \frac{v_i - \mu_{dset}}{\sigma_{dset}}$

Here, $\mu_{dset}$ and $\sigma_{dset}$ are the mean and standard deviation of $v_i$ for the full dataset.

### A.3 DATA SELECTION FLOPS

For FLOP calculation, we use the common approximation as performed by Kaplan et al. (2020) $C_{forward} = 2ND$ for forward pass and $C_{train} = 6ND$ for one finetuning pass which involves one forward and one backward pass, where $D$ denotes total number of training tokens, and $N$ is the number of parameters. Figure 5 shows a comparison between methods across models and datasets in terms of their average accuracy vs. FLOPs used during data selection.

| Dataset | Model | Total tokens |
|---|---|---|
| Social IQa | Llama 3 8B | 1,786,531 |
|  | OLMo 2 7B | 1,786,578 |
| CommonsenseQA | Llama 3 8B | 577,187 |
|  | OLMo 2 7B | 577,230 |
| CosmosQA | Llama 3 8B | 3,692,490 |
|  | OLMo 2 7B | 3,692,921 |

Table 4: Total tokens.

### A.4 IMPACT OF NUMBER OF CLUSTER ON OUR METHOD

In our method, we carry out clustering step over embeddings. In the main body of the paper, we utilize the heuristic $K = \frac{mn}{100}$ for all experiments. To validate this choice and assess the stability of our method, we conducted an ablation study varying $K$ across a relevant range when $m = 25$.

| Difficulty Score | Dataset | Data Selection FLOPs ($\times 10^{15}$) |
|---|---|---|
| Perplexity | Social IQa | 25 |
| | CommonsenseQA | 8.08 |
| | CosmosQA | 51.7 |
| Confidence | Social IQa | 225 |
| | CommonsenseQA | 72.7 |
| | CosmosQA | 465 |
| Variability | Social IQa | 225 |
| | CommonsenseQA | 72.7 |
| | CosmosQA | 465 |
| PVI | Social IQa | 450 |
| | CommonsenseQA | 145 |
| | CosmosQA | 931 |
| VoG | Social IQa | 225 |
| | CommonsenseQA | 72.7 |
| | CosmosQA | 465 |

Table 5: FLOPS utilized for data selection using Llama 3 8B.

| Difficulty Score | Dataset | Data Selection FLOPs ($\times 10^{15}$) |
|---|---|---|
| Perplexity | Social IQa | 28.6 |
| | CommonsenseQA | 9.23 |
| | CosmosQA | 59.1 |
| Confidence | Social IQa | 257 |
| | CommonsenseQA | 83.1 |
| | CosmosQA | 532 |
| Variability | Social IQa | 257 |
| | CommonsenseQA | 83.1 |
| | CosmosQA | 532 |
| PVI | Social IQa | 515 |
| | CommonsenseQA | 166 |
| | CosmosQA | 1060 |
| VoG | Social IQa | 257 |
| | CommonsenseQA | 83.1 |
| | CosmosQA | 532 |

Table 6: FLOPS utilized for data selection using OLMo 2 7B.

We tested values of $K$ values which are powers of two closest to and below the proposed heuristic $K = \frac{mn}{100}$. We evaluate the accuracy, across these variations. As can be seen in 6, our method's performance remains highly stable across the tested range of $K$ values.[anon]

## A.5 TRAINING HYPERPARAMETERS

We find the optimal values through hyperparameter search over learning rates {1e-05,2e-05}, batch size {64, 128} and, number of epochs {2,3}. Based on the performance, we set learning rate as 2e-05 and batch size as 128 and number of epochs as 3. Models are trained on selected subsets using the same hyperparameters that are used for training 100% of the data. This follows from previous works like (Maharana et al., 2023; Anand et al., 2023; Sorscher et al., 2022)[anon]

## A.6 ANALYSIS OF PERFORMANCE ACROSS DIFFICULTY INTERVALS

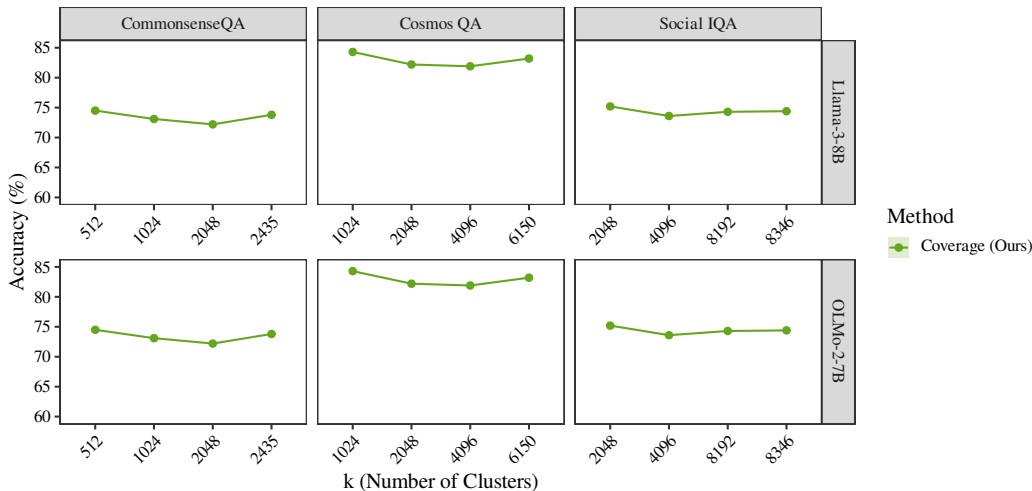

Figure 6: Per-dataset and average accuracy across different values of the clustering parameter $k$ at 25% selection.

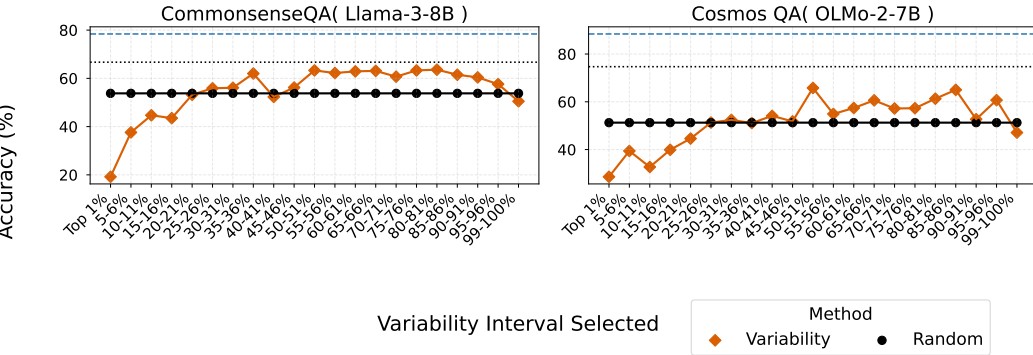

Figure 7: Comparative performance of 1% selected data on the Llama 3 8B and OLMo 2 7B models across CommonsenseQA and CosmosQA respectively, as a function of the interval used. For random selection, the points show the mean test accuracy. The blue dashed line indicates results from fine-tuning on 100% training data, while the gray dotted line indicates zero-shot performance.

A common concern when training LLMs on very small data budgets using the same hyperparameters used for full dataset is that the model may be undertrained. To verify that our training setup does not itself cause degradation in performance at small budgets, we perform the following diagnostic experiment. To examine this, we extended our analysis to interval-based difficulty selection, where samples of a fixed size (1%) are chosen from across the difficulty spectrum (e.g., 5–6%, 10–11%, up to 99–100%, the easiest examples). We consider two dataset–model combinations and examine how performance varies when selecting 1% subsets across the difficulty spectrum. Specifically, we first choose the top 1% most difficult examples, then 1% intervals such as 5–6%, 10–11%, and so on, up to the 99–100% examples (these being the easiest examples). For CosmosQA we fine-tune OLMo 2 7B, and for CommonsenseQA we fine-tune Llama 3 8B. We select these because, when using variability as the difficulty metric with 1% data selection, the observed performance is significantly worse than random sampling Fig 1. The results for these interval-based selections are compared with the random baseline. Fig 7 shows the performance when using 1% data from different intervals. These results support the view that the observed degradation for certain selection methods is not due to insufficient optimization or inappropriate hyperparameters. Instead, performance strongly depends on which data points are selected at small budgets.[anon]

## A.7 EFFECT OF LAYER ON CLUSTERING

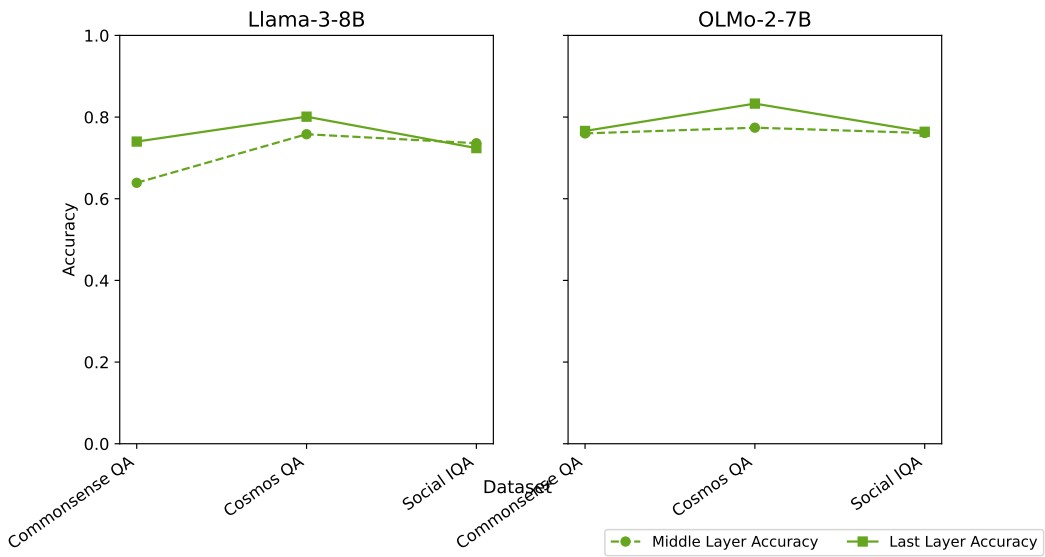

Figure 8: Comparative performance of 10% selected data on the Llama 3 8B and OLMo 2 7B models across all the datasetsAccuracy on CommonsenseQA, CosmosQA, and Social IQA using Last Layer vs. Middle Layer embeddings in the clustering step of our coverage based method.

In the main paper, we used the embeddings from the last layer for k-means in clustering in 7. To investigate if intermediate layer embeddings can be more helpful, we repeat our experiments using embeddings from the middle layer while selecting 10% of the data. Fig 8 shows the performance when using the last layer and the middle layer. Across models and evaluation benchmarks, intermediate-layer embeddings did not provide an improvement: final-layer embeddings consistently achieved higher average accuracy.[anon]

## A.8    MULTIPLE RUNS FOR HANDLING RANDOMNESS IN LLM FINETUNING

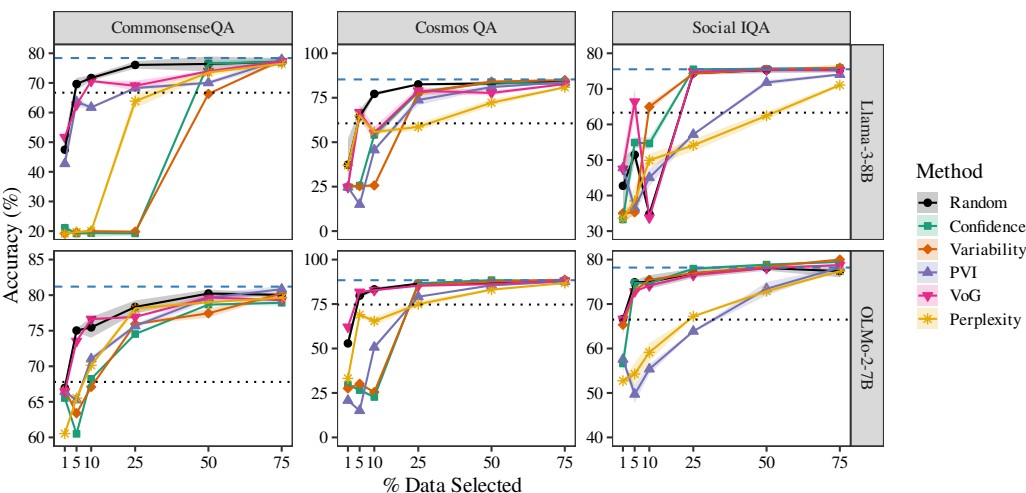

Figure 9: Comparative performance of difficulty-based data selection methods on the Llama 3 8B and OLMo 2 7B models across the three datasets, as a function of the percentage of training data selected. The points show the mean test accuracy and shaded regions indicate the $\pm 1$ standard deviation band across runs. The blue dashed line indicates results from fine-tuning on 100% training data, while the gray dotted line indicates zero-shot performance.

We repeat all our experiments with difficulty-based metrics for 3 random runs where the randomness mainly comes from the training. Fig 9 shows the performance across 3 random runs[anon]

