# OpenReview forum: "Rethinking Data Selection: The Importance of Coverage over Difficulty in Generative Fine-Tuning"
_ICLR.cc/2026/Conference — Submitted to ICLR 2026_

### Official Review · Reviewer_ELNV · 2025-10-17

**Soundness:** 1
**Presentation:** 3
**Contribution:** 1
**Rating:** 2
**Confidence:** 4

**Summary:**

The paper investigates how well the sample selection strategies that use the sample difficulty perform for training large language models trained for generative tasks (represented as multi-choice quest answering datasets). Overall, the paper finds these methods are outperformed by simple random selection and postulates that it is due to limited coverage of the training data distribution (i.e., the selected samples are representative only of a part of the distribution and cannot achieve strong performance). Based on this, the paper proposes to select samples based on diversity by proposing to use a simple clustering method.

**Strengths:**

The paper is well-written and easy to understand. The methods that are used in comparison are explained in detail.


Dealing with an interesting problem -- how to select a subset of samples that would lead to more efficient training of LLMs without sacrificing any performance

**Weaknesses:**

**Main finding is already well-known**

The main finding of the paper is that selecting samples based on diversity (i.e., to cover the whole training dataset distribution) is already well known and utilised in many different selection methods -- although they were used for typical supervised setting with "smaller models" (e.g., BERT fine-tuning) it makes sense the finding would transfer to the use of LLMs as well. An interesting question would be whether this is still the case in the true generative tasks -- such as translation, free-text responses, etc. -- the paper evaluates it only on multi-choice question answering, which is close to the typical supervised/classification setting. In addition, there are already different well-established strategies that utilise diversity for selecting samples (i.e., active learning). As such, I do not believe that the contribution of the paper is sufficient.

**Missing related work/selection strategies**

The paper focuses solely on selection strategies that utilise difficulty for the selection (or a single strategy that combines difficulty with diversity). The strategies are either already well established for LLMs and NLP, or are transferred from image or supervised classification domain. However, there are many existing strategies that are completely missing -- active learning strategies (that include selection based on diversity), core-set selection strategies (that try to select a subset that is the most representative of the full dataset) or other strategies that also implement simple clustering and and often combine different properties (although they may be used for in-context learning) [1-12] (to name a few).

Many of these strategies are well-established and often used (e.g., active learning) so I would expect a more extensive comparison that would include some of these strategies as well, especially when many of them already optimise for the coverage of the dataset and so are very similar to the strategy proposed in the paper.

There are also some inconsistencies in the paper in regards to the selection strategies. First of all, you claim that the Dataset Cartography paper suggests selecting hard samples for fine-tuning -- however, the main finding from the paper is that combination of easy to learn and ambiguous samples often works the best. Second, you argue that you do not consider Forgetting method because it would require large number of epochs -- however, it requires to run the same number of epochs as the Dataset Cartography (and other approaches that utilise training dynamics). I would agree that defining what "forgetting" means for generative tasks (i.e., open ended tasks with free text answers where multiple answers might be correct), but you do not work in such setting -- in my opinion multi-choice QA is more or less a classification tasks as the model is presented with 4+ options and just selects one of them.

**Questionable experimental setup**

The experimental setup for training the LLMs is a bit questionable for me, although there is not many details included (batch size?). For example, the LLMs are trained only for 3 epochs, which in my experience is often insufficient, especially when the most difficult samples are selected. This is further reinforced by the results show in the paper, where often training the LLM with less than 25% of dataset leads to performance that is lower than the same model used without training (which for me indicates specific problems in the training process). At the same time, the exact same setup is used for subsets of different sizes, irrespective of whether 100 samples are used for training, 1000 or 10000. In each setting, the selection of these hyperparameters (combination of epochs and batch size) would lead to completely different models (sometimes undertrained, sometimes overtrained). I would suggest dedicating more time to finding an optimal set of parameters, especially considering that previous works found that selection methods outperform random selection on smaller sizes, but later all of them behave similarly to random selection.

Furthermore, for most of the methods, results from only a single run are presented. However, LLMs were observed to be especially affected by the randomness in training (which is most significant when using low number of samples), which can also be seen in the results, and so this setup could lead to biased results. I would suggest repeating the training multiple times and report the deviation from different samples as that would provide more comprehensive evaluation.

Overall, I believe that many of the findings in the paper can be explained by the experimental setup leading to undertrained models, and if fixed would lead to different findings.

**References**
1. A Survey of Deep Active Learning
1. Deepcore: A comprehensive library for coreset selection in deep learning
1. Datamodels: Predicting Predictions from Training Data
1. Data Curation Alone Can Stabilize In-context Learning
1. Dataset Cartography: Mapping and Diagnosing Datasets with Training Dynamics
1. Cartography Active Learning
1. Finding Support Examples for In-Context Learning
1. Automatic Combination of Sample Selection Strategies for Few-Shot Learning
1. EXPLORA: Efficient Exemplar Subset Selection for Complex Reasoning
1. Sample Efficient Demonstration Selection for In-Context Learning
1. MEAL: Stable and Active Learning for Few-Shot Prompting
1. On Training Instance Selection for Few-Shot Neural Text Generation

**Questions:**

What specific hyperparameters were used for the different models and training subsets?


How does the comparison between methods look like on true generative tasks (translation, summarisation, free-form question answering)?

---

> ### Author Response · Authors · 2025-11-27
> **Response to Review**
>
> We thank you for your time, effort, and insightful comments. We are glad you found our paper to be well-written, and that we are dealing with an interesting problem. Please see our response below:
>
> **Transfer of findings from previous works**: We agree that these techniques have been applied in small model classification tasks for NLP. But it is not clear how well they transfer to LLM generative fine-tuning. This is observable from the low performance of difficulty-based methods in our setting.
>
> **Experiments on tasks more generative in nature**: Please see general response
>
> **Comparison with Active Learning (AL) literature**: We agree that our coverage based method is similar in spirit to diversity enhancing methods in the AL literature. However, our methods differ in their goals. While our work aims to understand why data selection based methods based on difficulty/uncertainty underperform in a generative fine-tuning setting, AL’s goal is largely for labeling cost reduction. AL requires an iterative loop: repeated, costly inference on the entire unlabeled pool and multiple retraining steps. For LLMs, where both inference and training are prohibitively expensive, this iterative approach is impractical. We apply the coverage based method for one-shot static data selection. Specifically, we establish that success in this setting is critically dependent on maximizing data coverage/diversity. We show that the connection between the difficulty based methods and high data coverage is weak (explaining why difficulty based methods  fail). We use the superior performance of coverage based methods as evidence to validate that coverage is the critical factor that must be prioritized over difficulty metrics for LLM fine-tuning.
>
> **Missing related work/selection strategies**:We would like to thank the reviewer for putting together a detailed list of the works relevant to ours. We will add this to the discussion in the related work section. We maintain while these works are thematically relevant, they address substantially different problems.
>
> DataModels [Ref 3 in review]: A limitation of using datamodels for data selection is that it implicitly assumes that the influence of selected data can be added linearly (i.e., two equally scored data points are expected to doubly improve the model performance ). This assumption does not hold in practice e.g., adding redundant data does not help performance [1, 2, 6]. Our analysis highlights the importance of diversity.
>
> [Ref 4,7,10 in review] deals with data selection in In-context learning. But we focus on data selection in task specific generative fine-tuning. ICL is fundamentally different from our setting because it deals with selection at inference time. Similarly, [Ref 8, 11, 12 in review] deals with few-shot finetuning.
>
> [Ref 6 in review] focuses on data selection in active learning as pointed out is different from our one-shot static data selection.
>
> **Inconsistencies about data cartography**: We do not claim that the cartography paper advocates favoring only the hard examples and instead treat their metrics to define a notion of hardness since the goal in our first experiment is to analyze how well difficulty-based scores fare. We will clarify this in the paper.
>
> **Forgetting Score**: Thanks for pointing it out. Meaningful forgetting scores require evaluating the entire training set to detect these flip events.To obtain non-obvious, meaningful forgetting scores during the rapid convergence of fine-tuning, the entire training set must be evaluated extremely frequently (every minibatch). This frequent, full-set evaluation is what makes the procedure computationally prohibitive in our LLM fine-tuning scenario. We will clarify the justification in our paper.
>
> **Questionable experimental setup**:We have attempted to address this query in the general response, where we expand on why we fixed the number of epochs and demonstrate how our results cannot simply be explained by the presence of a “fundamental flaw” in the training runs/setup and instead by data quality.
>
> **More runs**: We have repeated our runs on all difficulty-based metric experiments using the LLaMA model across our three datasets at 1%, 5%, 10% and 25% selection. The results from these repeated runs do not show any change in the relative ranking of methods compared to what we initially reported. We will report the results again as soon as our experiments are completed. We have included the partial results in appendix  A.8  in the revised paper.
>
> **Hyperparameters**: We use the same hyperparameters as specified in the section 4. We added additional detail in section A.5 in the Appendix
>
> **Comparison with other generative task**: Please see general response
>
> References:
> [1] Most influential subset selection: Challenges, promises, and beyond.
> [2] Simfluence: Modeling the Influence of Individual Training Examples by Simulating Training Runs
> [3] Understanding goal-oriented active learning via influence functions

---

> > ### Author Response · Authors · 2025-12-03
> > **Update to Reviewer ELNV**
> >
> > Dear Reviewer ELNV,
> > We have completed the additional runs across all datasets and models, now including 50% and 75% selection in addition to the previously reported 1%, 5%, 10%, and 25%. The full results have been incorporated into the revised paper (Appendix A.8). The new results confirm our original conclusions, with no change in the relative ranking of methods. Thank you again for your thorough feedback

---

### Official Review · Reviewer_nNqv · 2025-10-30

**Soundness:** 2
**Presentation:** 3
**Contribution:** 2
**Rating:** 2
**Confidence:** 4

**Summary:**

This paper proposes a clustering-based data selection method for LLM generative finetuning that samples one instance from each cluster in a partitioned embedding space. While this method does not consistently outperform random selection, it demonstrates more stable, monotonically improving performance and robustly outperforms existing difficulty-based methods.

**Strengths:**

-	Compelling analysis and insight: The benchmarking and analysis of using difficulty-based methods for LLM generative fine-tuning provide valuable insights and well support the motivation of this work.
-	The proposed coverage-based method presents reasonable performance improvements over existing difficulty-based methods.

**Weaknesses:**

-	Unclear empirical advantage of the proposed method:  The primary weakness is that the proposed coverage-based method, while more stable, does not provide a clear and consistent performance improvement over the random selection baseline. It remains on par with random selection. This raises the question of why not just use random selection, which is simple, efficient and effective.
-	Limited technical contribution: The proposed method simply uses K-means to cluster the training data and selects one sample from each cluster, which is trivial. It does not introduce new techniques for data selection. In my opinion, the authors should try to improve their current method to achieve better performance, at least better than random selection.
-	Narrow model scopes and tasks: The proposed method is only evaluated on Llama 3 8B and OLMo 2 7B, which is limited. The performance on larger models (e.g., 13B and 32B) and other pretrained backbones (e.g., Qwen) is unclear. Furthermore, the experiments cover only multiple-choice QA tasks. Evaluation on a broader range of generation tasks, such as instruction-following and mathematics tasks, can further verify the effectiveness of the proposed method.
-	Clustering hyperparameter sensitivity: The proposed method's performance is likely sensitive to the choice of the number of clusters \$k\$. The paper fixes \$k = mn/100\$, but this is a heuristic. A more thorough ablation or justification for this choice would strengthen the soundness of this paper.

**Questions:**

See the Weaknesses. The authors should address these limitations.

---

> ### Author Response · Authors · 2025-11-27
> **Response to Review**
>
> We thank you for your time, effort, and insightful comments. Please see our response below:
>
> **Addressing empirical advantage and technical contribution of the proposed method**: We agree with the reviewer that random selection is a strong baseline but we must clarify that **our primary contribution** is about recognizing a relationship between coverage and catastrophic accuracy drop that is observed with existing difficulty based techniques for data selection for LLM generative finetuning. Our introduction of a distribution-based notion of coverage is to further demonstrate that difficulty-based methods fundamentally overlook coverage, which explains their poor and unstable performance. Furthermore, **our notion of coverage provides the explanation for the exceptional performance of random sampling and a reason for why difficulty-based methods might fail.**
>
> **Narrow model scopes and tasks**: While at face value we have only focused on two models and 3 tasks, our total experiments amount to 438 final fine-tuning runs of models in the 7B to 8B parameter range [12 (5 difficulty + 3 random + 3 coverage + 1 D2) * 6 (percentages) * 3 (datasets) * 2 (models) + 6 (100% runs)), which is quite significant for resources available at academic institutions. Furthermore, our task selection was not random but rather motivated by observing significant differences between zero-shot and fine-tuning performance, which we have mentioned in section 4 of the original draft.
>
> **Evaluation on a broader range of generation tasks**: Please see general response.
>
> **Clustering hyperparameter sensitivity:**  Thank you for the suggestion. We run experiments along these lines by varying K. We  employ a round-robin method whereby we iterate through the clusters, selecting one example at random from each cluster. This process is repeated until K examples have been selected. Our results are shown in Figure 6, Sec. A.4 in the Appendix. Our experiments confirm that results remain stable across these choices, indicating that our method is not sensitive to the precise value of K.

---

### Official Review · Reviewer_oRAC · 2025-10-31

**Soundness:** 3
**Presentation:** 3
**Contribution:** 3
**Rating:** 6
**Confidence:** 3

**Summary:**

This paper (1) evaluates some difficulty-based data point selection methods for their utility in selecting small portions of multiple-choice datasets for finetuning language models, (2) shows that the performance of these methods doesn’t beat random sampling, (3) shows that the performance of these methods correlates reasonably well with an intuitive notion of how much their resulting data subsets cover the distribution defined by the full finetuning dataset, and (4) demonstrate that a coverage-based dataset selection method works roughly as well as random sampling, with potentially lower variance.

**Strengths:**

The MAUVE-inspired coverage metric proposed in this paper is nice and thought-provoking.
I find the experiments well-documented, reasonably convincing for the claims made in the paper (though, see the weaknesses section.) I think that the superiority of random sampling as a subset selection strategy is a nice result, and the notion of coverage over difficulty being important is again thought-provoking.

I think with additional caveats in the introduction that mention how we’re only working with multiple-choice finetuning problems, this paper provides a nice, well-scoped result that demonstrates both the efficacy of random sampling as dataset selection over difficulty-based methods, and how a coverage-based method also does just about as well as random sampling.

**Weaknesses:**

The paper claims that the generative case of LMs makes for a difficult new aspect to dataset selection, but it only trains and evaluates on multiple-choice question datasets. This means the claim that the LM “retains its generative capabilities” is also questionable – if you finetune enough on multiple-choice datasets, models do not retain (meaningful) generative capabilities, despite technically still defining distributions over strings. I was surprised by the difference between the intro language (focus on generative aspects of LMs) and the experimental setting (multi-choice, even when parameterized through the likelihood of the choices under the LM’s distribution.)

Overall, I’m not sure why we care about finetuning efficiency in pretrained language models and I’d like the authors to motivate this better. Almost all of the FLOPs that go into the resulting finetuned model have already happened – during pretraining. Even though in the modern era RL is consuming a larger and larger fraction of the flops compared to pretraining, this paper doesn’t consider that setting. My understanding is that the datasets considered are pretty tiny – <4M tokens according to table 4 – so the benefits of using only 1% of that data are unclear to me. If we care about, e.g., sample efficiency – as in, the cost is not in the FLOPs in the model but instead in gathering the data – then the experimental setting should change to independently hyperparameter-optimize for each number of samples (instead of fixing the number of epochs, for example.) Also, the motivation provided by the authors for data selection for breaking scaling laws — citing Sorscher et al., 2023 is indeed for pretraining (or large-scale training in general.)

**Questions:**

- Why fix the number of epochs in training on a number of samples (instead of more epochs for the smaller sample counts?) I realize this may be to have fewer samples mean less compute going into finetuning, but the real cost of finetuning examples seems to be in getting them, not training on them. Even so, one could increase the learning rate in the few-sample case instead to try to learn more from them in the same amount of compute!
- Footnote 3 –  “In practice, we use the model’s hidden state representation at the last layer and last position as our embedding extractor.” I’m not considering this a weakness to be clear, but I can’t help myself saying that I recommend not doing this! This last layer is a really lexically-driven vector because it must be a low-rank representation of the distribution over the next word. Vectors from the middle of the network would probably work much better for a nice clustering! If you tried this already and it works worse, I’d love to know.

---

> ### Author Response · Authors · 2025-11-27
> **Response to Review**
>
> We thank you for the comments and positive feedback! Please find our responses
>
> **Evaluation only on MCQ**: Please see general response
>
> **Motivation for fine tuning efficiency**: We agree with the reviewer that compared to pre-training FLOPs, the computational cost of fine-tuning is small. However, data selection for fine tuning can be helpful when dealing with specialized domains. While LLMs obtained through pre-training excel in general language tasks, they may not deliver optimal outcomes in specialized domains, such as medicine, or finance. To maximize performance in specialized domains, models fine-tuned on domain data offer superior capabilities over LLMs. Moreover, as mentioned in our paper, data selection has also shown to be helpful in instruction tuning [1] and reasoning [2].
>
> **Vary the number of epochs**: We have attempted to address this query in the general response, where we expand on why we fixed the number of epochs and demonstrate how our results cannot simply be explained by the presence of a "fundamental flaw" in the training runs/setup and instead by data quality.
> In our setup, we focus on reducing training time and using less compute. This is particularly beneficial for large models that require high computation for training. If we were to expend the same amount of compute  on a smaller subset of the data as the full dataset by increasing the number of epochs, then we will incur the high computation cost that we are trying to avoid by doing data selection. Our approach of maintaining fixed hyperparameters across all budget sizes is followed in the literature (e.g., Refs. [3, 4, 5]), which focuses on relative gains. We confirm that we swept through hyperparameters and found that the selected values work well regardless of data budget. Please refer to general response for more details.
>
> **Vectors from the middle of the network**: We chose the final hidden layer representations as this is the standard practice in much of the coreset and representation-based data selection literature. Thank you for suggesting the experiment. Our initial exploration using a middle layer at 10% subset selection gave lower performance on the downstream tasks. Please look at section A.7 in the appendix
>
> [1] LIMA: Less is more for alignment.
>
> [2] Limo: Less is more for reasoning
>
> [3] D2 Pruning: Message Passing for Balancing Diversity and Difficulty in Data Pruning
>
> [4] Beyond neural scaling laws: beating power law scaling via data pruning
>
> [5] Influence Scores at Scale for Efficient Language Data Sampling

---

### Official Review · Reviewer_2D8L · 2025-11-03

**Soundness:** 3
**Presentation:** 2
**Contribution:** 2
**Rating:** 6
**Confidence:** 3

**Summary:**

This paper explores different data selection methods for fine-tuning LLMs and comparing against difficulty based metrics such as Perplexity, Confidence, V-usable information, etc.). This paper defines the notion of coverage as how well a data selection represents the entire distribution of the training set. A key idea of this work is that coverage and a random selection baseline outperform difficulty-based metrics in improving model performance during fine-tuning.  The authors argue that diversity in training data is more important than difficulty scores.

**Strengths:**

1. The paper argues, with empirical evidence, that coverage (the representation of the entire data distribution) is more important for generative tasks like LLM fine-tuning than relying on difficulty-based metrics. It shows that coverage-based methods outperform methods based on difficulty (e.g., Perplexity, Confidence) in terms of fine-tuning model performance.
2. The experiments are well-executed with careful consideration of removing randomness in data selection and repeated trials - particularly time-consuming with limited resources and larger models.
3. Usage of k-means clustering of model embeddings offers a new and computationally efficient means of data selection in LLMs. It avoids re-training and reduces compute overhead.

**Weaknesses:**

1. This paper proposes the random baseline as a rightly powerful one. However, it would significantly benefit from comparisons with data-attribution methods [1] that exist with the newly proposed coverage-based approach. It will be interesting to see how coverage compares with existing dataset selection methods (semi-value-based or influence-function-based) for fine-tuning LLMs / generative tasks that implicitly or explicitly use distributional coverage - such as [2], [3], [4] etc. and when it could have a distinct advantage.



[1] A Survey of Data Attribution: Methods, Applications, and Evaluation in the Era of Generative AI
[2] Get more for less: Principled Data Selection for Warming Up Fine-Tuning in LLMs
[3] What is Your Data Worth to GPT? LLM-Scale Data Valuation with Influence Functions
[4] DsDm: Model-Aware Dataset Selection with Datamodels

**Questions:**

1. Can the authors comment on the computational time complexity of this k-means clustering approach - at what point is it faster to train on a large dataset than a careful selection (since LLMs can generalize over the course of training with enough data).
2. Could the authors offer light on when their method is likely to have an advantage over the existing semi-value-based or influence-function-based dataset selection paradigms.
3. If data quality is low to begin with, can this approach sift out clean and high-quality data subsets (which could be a huge contribution). For instance, noisy data or bad quality data can bias coverage towards the long-tailed poor quality samples.

---

> ### Author Response · Authors · 2025-11-27
> **Response to Review**
>
> We thank you for the comments and positive feedback! We also thank the reviewer for pointing out the connection between data valuation and data selection.
>
> **Comparison with influence-based methods**: Influence function-based approaches are not directly applicable to the problem we address. These methods rely on the availability of a validation set(target set) that is a very good representative of the test set, as they find training examples (from candidate dataset ) most similar to the validation examples. This is a different problem from ours, which does not assume the availability of a good validation set.
>
> **Computational complexity**: The computational complexity of our approach can be divided into the embedding calculation and clustering steps. The embedding calculation step involves a forward pass through the model as so the total computation cost will be 2*N*D FLOPS where N is the number of model parameters and D is the dataset size. The clustering step will take O(D*k*I*d) where D is the dataset size, k is the number of clusters, I is the number of iterations and d is the hidden layer dimensionality. The cost is largely dominated by the embedding calculation steps
>
> **Faster to train on larger dataset**: While we cannot definitively answer without additional investigation, we can comment that at low compute budgets selecting high-quality data seems clearly beneficial. However, when compute far exceeds the available data, repeatedly training on the same high-quality subset might yield diminishing returns. In this regime, lower-quality data can become more useful than reusing the saturated high-quality data.
>
> **Advantage over influence based and semi-value-based**:
> A major limitation of using influence functions for data selection is that they implicitly assume that the influence of selected data adds linearly (i.e., two equally scored data points are expected to doubly improve the model performance [1,3,4]). This assumption does not hold in practice e.g., adding redundant data does not help performance. Our method avoids this issue by explicitly prioritizing coverage over difficulty, ensuring that selected data are diverse and non-redundant.
> Semi-value-based approaches (e.g., Shapley/Banzhaf variants) conceptually capture interactions among data points, but these approaches require evaluating all possible subsets of the dataset, making it computationally infeasible for data selection in training LLMs. By contrast, our method requires only a single forward pass for embeddings and a clustering step, while still capturing the key effect of coverage.
>
> **Sift high-quality data**: We note that our current work focuses on the idealized clean-data setting, but data selection under noisy data is an interesting extension. We concur that the goal of coverage is to select a subset that mimics the entire data distribution, and applying selection based purely on coverage without a quality measure would be insufficient in low quality data settings.
> A natural way to address this is to first identify a simple metric, such as model confidence, loss, or embedding-based outlier scores to identify low quality samples. Prior work (e.g.[2]) shows that low-confidence samples often correlate with poor quality or noisy labels. We can modify the selection strategy so that instead of selecting a point randomly we can select points with high quality (e.g., high confidence). This ensures that the selected data instance covers that part of the distribution and is not a noisy data point, thereby achieving coverage.
>
> [1] Understanding Goal-Oriented Active Learning via Influence Functions
>
> [2] Dataset Cartography: Mapping and Diagnosing Datasets with Training Dynamics
>
> [3] Most influential subset selection: Challenges, promises, and beyond.
>
> [4] Simfluence: Modeling the Influence of Individual Training Examples by Simulating Training Runs

---

### Author Response · Authors · 2025-11-27
**General Response to Reviewers**

Dear reviewers,

We thank you for your time, effort, and insightful comments. The feedback has improved the experiments and analysis in our paper (see revised pdf; new text is in magenta). Here, we address concerns raised by more than one reviewer and point to sections in the revised PDF wherever applicable. For other concerns, please see individual responses.

**Generative Fine-Tuning for MCQA  (raised by ELNV, nNqV, and oRAC)**: We choose MCQA to ensure rigorous, automated evaluation. This setting allows us to isolate the efficacy of our dataset selection method by controlling for the noise and subjective scoring inherent in open-ended generation (hallucination, stylistic variation). Importantly, although our evaluation uses MCQA datasets, our fine-tuning setup is fully generative: we do not introduce a classification head or a discriminative objective. Instead, we fine tune using the next token prediction loss as done in the instruction tuning literature. Our training target is not a single label index but a sequence as discussed in paragraph “Difficulty score computation” in section 4. We expect that the **generative nature** of our fine-tuning setup is the key reason driving the substantial difference between our difficulty-based methods findings and those reported in classification-based settings.

We used benchmarks where we see substantial performance gaps between finetuning and zero-shot setting as mentioned in Section 4 of the original draft. This indicates that generative finetuning setup is reasonable. Finally, the evaluation involves generating the response and checking for exact match, rather than calculating logits over pre-defined classes.

**Potential issues with experimental setup (raised by ELNV and oRAC)**: Since our experiments involved 438 final fine-tuning runs of models in the 7B to 8B parameter range [12 (5 difficulty + 3 random + 3 coverage + 1 D2) * 6 (percentages) * 3 (datasets) * 2 (models) + 6 (100% runs)), we did  hyperparameter search for number of epochs over {2,3} as tried  in SFT stage of OLMo [4] . On the topic of hyperparameter tuning, we have spent an equivalent amount of time tuning hyperparameters as the works we borrow from, and that this approach also follows from previous work  [1, 2, 3] fixing the hyperparameters across budget sizes. Nonetheless, we swept through a bunch of hyperparameters initially, finding that the current ones as specified in section A.5 in the Appendix yielded the best performance.

In the context of training problem vs. data-selection problem, if our results were simply explained by the fact that there is something wrong with the amount of training or the training runs themselves, then we should see little difference between results when selecting different batches of N% of data (especially when N is small). To test this hypothesis, we conducted additional experiments where we selected 1% samples at different areas of the difficulty spectrum ranging from the top 1% most difficult (acc. to variability) all the way to the top 1% most “easiest” samples at intervals of 5-percent points. We did this for Llama3 on Commonsense QA and OLMo 2 on Cosmos QA since these were where these models struggled the most at 1%. Our results are shown in **section A.6 in Appendix of the updated draft**. We see that the models’ accuracy significantly increases at different batches of 1% samples, suggesting that our results cannot be explained by fundamental flaws in the training run but rather it really being a data-selection problem. These results also further showcase the empirical failures of difficulty-based selection strategies, reinforcing our conclusions.

[1] D2 Pruning: Message Passing for Balancing Diversity and Difficulty in Data Pruning

[2] Beyond neural scaling laws: beating power law scaling via data pruning

[3] Influence Scores at Scale for Efficient Language Data Sampling

[4] 2 OLMo 2 Furious

---

### Meta-Review · Area_Chair_YcT6 · 2025-12-31

**Summary:**

This paper studies data selection strategies for generative fine-tuning of large language models and argues that coverage of the data distribution is more important than difficulty-based selection. Several reviewers found the emperical evaluation of the paper provides insights and aligns with the claim of the paper. However, after the rebuttal, several key concerns remain insufficiently addressed, including the motivation for efficient fine-tuning, whether the observations generalize to larger-scale models, and the distinction between classification-style evaluations and generative tasks. The latter also raises concerns about the paper’s novelty, as similar coverage- or diversity-based selection strategies have been explored extensively in prior NLP classification settings.

Considering all these factors, the Area Chair recommends rejection and encourages the authors to incorporate the reviewers’ suggestions when submitting to a future venue.

**Reviewer Concerns:**

The reviewers have the following major concerns:
1. It remains unclear whether the MCQA setting should be considered a generative or a classification task. In the rebuttal, the authors argue that their fine-tuning procedure follows a generative formulation and should therefore be treated as a generation task. However, the Area Chair finds this justification insufficient and believes that evaluations on open-ended generation tasks would be necessary to more convincingly support the paper’s arguments.

2. The motivation for efficient fine-tuning is not sufficiently clear. In the rebuttal, the authors acknowledge that the computational cost of fine-tuning is relatively low, and argue that data selection may be particularly beneficial for domain-specific tasks such as finance and healthcare, as well as for instruction tuning and reasoning. However, the Area Chair finds this argument insufficiently supported, as it is not backed by empirical evidence. It therefore remains unclear how well the proposed method would generalize to these settings.

3. It remains unclear whether the conclusions of the paper generalize to larger models. In the rebuttal, the authors note that they lack sufficient computational resources to scale up the experiments, which does not directly address this concern.

4. As a result of the first concern, the novelty of this paper is difficult to justify relative to data selection methods previously explored in NLP classification tasks.

The authors addressed several other concerns, including comparisons with influence-based methods, sensitivity to clustering hyperparameters, and other minor issues.

**Reviewer Scores:**

All reviewers would keep their scores, as the rebuttal did not resolve the core concerns raised by the reviewers.

---

### Decision · Program_Chairs · 2026-01-26

Reject